# Theophylline-Induced Relaxation Is Enhanced after Testosterone Treatment via Increased K_V_1.2 and K_V_1.5 Protein Expression in Guinea Pig Tracheal Smooth Muscle

**DOI:** 10.3390/ijms24065884

**Published:** 2023-03-20

**Authors:** Jorge Reyes-García, Verónica Díaz-Hernández, Abril Carbajal-García, María F. Casas-Hernández, Bettina Sommer, Luis M. Montaño

**Affiliations:** 1Departamento de Farmacología, Facultad de Medicina, Universidad Nacional Autónoma de México, Mexico City 04510, Mexico; 2Departamento de Embriología y Genética, Facultad de Medicina, Universidad Nacional Autónoma de México, Mexico City 04510, Mexico; 3Laboratorio de Hiperreactividad Bronquial, Instituto Nacional de Enfermedades Respiratorias “Ismael Cosío Villegas”, Mexico City 14080, Mexico

**Keywords:** testosterone, airway smooth muscle, theophylline, delayed rectifier K^+^ channels

## Abstract

Theophylline is a drug commonly used to treat asthma due to its anti-inflammatory and bronchodilatory properties. Testosterone (TES) has been suggested to reduce the severity of asthma symptoms. This condition affects boys more than girls in childhood, and this ratio reverses at puberty. We reported that guinea pig tracheal tissue chronic exposure to TES increases the expression of β_2_-adrenoreceptors and enhances salbutamol-induced K^+^ currents (IK^+^). Herein, we investigated whether the upregulation of K^+^ channels can enhance the relaxation response to methylxanthines, including theophylline. Chronic incubation of guinea pig tracheas with TES (40 nM, 48 h) enhanced the relaxation induced by caffeine, isobutylmethylxanthine, and theophylline, an effect that was abolished by tetraethylammonium. In tracheal myocytes, chronic incubation with TES increased theophylline-induced IK^+^; flutamide reversed this effect. The increase in IK^+^ was blocked by 4-aminopyridine by ~82%, whereas iberiotoxin reduced IK^+^ by ~17%. Immunofluorescence studies showed that chronic TES exposure increased the expression of K_V_1.2 and K_V_1.5 in airway smooth muscle (ASM). In conclusion, chronic exposure to TES in guinea pig ASM promotes upregulation of K_V_1.2 and K_V_1.5 and enhances theophylline relaxation response. Therefore, gender should be considered when prescribing methylxanthines, as teenage boys and males are likely to respond better than females.

## 1. Introduction

Theophylline, a methylxanthine naturally occurring in small amounts in tea leaves, cocoa, and coffee seeds [1], has been used for decades to treat asthma and chronic obstructive pulmonary disease (COPD) [2]. Because of its anti-inflammatory and bronchodilator properties, this methylxanthine is an important therapeutic agent in patients with poorly controlled asthma who use inhaled corticosteroids and β_2_-adrenergic agonists, or in patients who do not have access to these medications [2,3,4,5]. The bronchodilator effect of theophylline (1,3-dimethylxanthine) was first described by Macht and Ting [6] in bronchial pig preparations in 1921. Later, in 1922, an optimistic response to the administration of theophylline in combination with theobromine to asthmatic subjects was reported [7]. However, clinical assays of theophylline’s palliative effects and improvement of pulmonary function in asthmatics were not conducted until the 1930s [7,8], and the drug was approved by the FDA in 1940 [9]. In 1973, Mitenko et al. [10] reported that theophylline plasma concentrations of 5–20 mg/L produced a dose-dependent improvement in pulmonary function in humans. Nevertheless, Turner-Warwick [11] found that a minimum concentration of 10 mg/L was required to relieve symptoms in asthma patients. Therefore, the therapeutic interval for theophylline plasma concentrations was set at 10–20 mg/L (~55–110 μM) [2], with the occurrence of side effects above the recommended maximum. Consequently, the use of sustained-release theophylline tablets was established to avoid such side effects [12,13]. In addition, other natural methylxanthines, including caffeine and theobromine, exert relaxing effects on airway smooth muscle (ASM) and are thought to improve airway function [14,15].

Theophylline is a non-selective inhibitor of cyclic nucleotide phosphodiesterase (PDE) isoenzymes [16]. Eleven different families (PDE1-PDE11) of ~24 PDE isoforms are responsible for the degradation of cyclic adenosine monophosphate (cAMP) and cyclic guanosine monophosphate (cGMP) [16,17]. In ASM, PDE4, and PDE3 in minor contribution, represent most of the cAMP-hydrolyzing PDEs [18,19,20,21]. Inhibition of PDEs by theophylline increases intracellular cAMP levels and causes bronchial relaxation by several mechanisms, which include activation of cAMP-dependent protein kinase A (PKA) [18,22]. This enzyme is a serine/threonine kinase with multiple targets in smooth muscle cells. PKA phosphorylates phospholipase Cβ (PLC_β_) and inositol triphosphate (IP_3_) receptors (IP_3_Rs), inhibiting smooth muscle contraction mediated by PLC_β_-IP_3_ signaling [23,24,25,26]. Additionally, phosphorylation of phospholamban (PLB) by PKA abolishes the inhibition of sarcoplasmic reticulum (SR) Ca^2+^-ATPase (SERCA), promoting Ca^2+^ uptake [27]. Moreover, PKA-dependent phosphorylation of high conductance Ca^2+^-activated K^+^ (BK_Ca_, also known as K_Ca_1.1) channels [28,29] and voltage-dependent rectifier K^+^ (K_V_) channels [30,31,32] increases their opening probability. Among the K_V_ channels, K_V_1.2, K_V_1.5, and K_V_7.5 appear to be the most relevant subtypes in ASM [32,33,34,35]. BK_Ca_ and K_V_ channels play an important role in airway smooth muscle relaxation induced by agonists such as ATP, serotonin, or the β_2_-adrenoreceptor (β_2_-AR) agonist salbutamol, which is widely used in asthma treatment [36,37,38]. Furthermore, the blockade of BK_Ca_ channels reduces theophylline-induced human airway relaxation, suggesting that this methylxanthine probably induces the opening of these channels via a PKA-dependent mechanism [39].

Androgens such as testosterone (TES) have been associated with lower symptom severity and physiopathology in male asthma patients [40,41,42]. During childhood, boys are more prone to asthma symptoms compared to girls. However, during puberty, the risk of developing asthma symptoms decreases in teenage boys [43,44]. The different patterns of asthma occurrence in males may be related to plasma TES levels. In males aged 13 to 80 years, plasma concentrations of this sex hormone range from 6 to 50 nM and reach their maximum between puberty and early adulthood [45,46]. Androgens exert their physiological effects through nongenomic and genomic pathways [47]. The nongenomic effects usually occur within seconds to minutes and do not involve androgen receptor activation (AR) or genomic transcription and protein synthesis. Therefore, these effects are not blocked by androgen receptor antagonists or transcription inhibitors [47,48,49,50]. Several works have shown that most of the nongenomic effects of TES in ASM are associated with the blockade of Ca^2+^ channels in the plasma membrane or in the SR [50,51,52,53,54]. In addition, TES is thought to impair the action of catechol-O-methyl transferase (COMT) and extra-neuronal uptake_2_ [55,56,57]. All these nongenomic effects are associated with ASM relaxation. In contrast, genomic effects are mediated by androgen-dependent stimulation of AR located in the cytoplasm, leading to genomic transcription and protein synthesis. However, these processes have been poorly investigated in ASM, and most studies addressing the effects of TES on the airways focus on inflammation. For example, innate group 2 lymphoid cells (ILC2) and CD4+ Th2 cells (Th2), which are involved in the type 2 inflammation characteristic of asthmatic airways, are known to be negatively regulated by TES [58,59]. Moreover, TES also reduces the expression of IL-17A, neutrophilic inflammation, and mucus production observed in some asthma patients [58].

Recently, we found that chronic ASM exposure to a physiological TES concentration (40 nM) for 48 h enhanced salbutamol-induced relaxation via increased expression of the β_2_-AR, a genomic effect that was blocked by flutamide (Flu) an antagonist of the AR [37]. In the same work, we showed that chronic TES treatment of ASM cells increased K_V_ and BK_Ca_-mediated outward currents triggered by salbutamol [37]. In this context, other authors have examined the nongenomic and genomic effects of TES on BK_Ca_ and K_V_ channels. For instance, Deenadayalu and colleagues [60] showed that micromolar concentrations of TES directly activated BK_Ca_ channels in porcine coronary artery myocytes and relaxed prostaglandin F2α (PGF_2α_)-precontracted coronary artery smooth muscle. Furthermore, Han and colleagues [61] demonstrated that nanomolar concentrations of TES increased BK_Ca_ unitary currents in human corporal smooth muscle cells. With respect to genomic effects, castration of male rats reduces K_V_7.1 expression in cardiomyocytes, and supplementation with TES improves the expression of this channel after two weeks [62].

Given that K_V_ and BK_Ca_ channels are downstream targets in the signaling pathway of cAMP-PKA and that previous studies by us and others have shown TES to induce genomic and nongenomic effects on these channels, we hypothesized that this androgen may enhance theophylline-elicited bronchorelaxation due to the upregulation of K^+^ channels. Therefore, we conducted organ bath experiments to investigate the effect of TES on theophylline-induced guinea pig ASM relaxation. We also evaluated theophylline-induced K^+^ currents in cultured myocytes and the expression of K^+^ channels in guinea pig ASM.

## 2. Results

### 2.1. Testosterone Enhances Tracheal Smooth Muscle Relaxation Induced by the Methylxanthines Caffeine and Isobutylmethylxanthine in Association with K^+^ Channels

Recently, we showed that storage of tracheas at 9 °C for 48 h does not alter the smooth muscle contractile apparatus [37]. Stimulation with the caffeine of tracheal rings precontracted with histamine (10 μM) (sham control) resulted in a concentration-dependent relaxation. In contrast, when tissues were treated with TES 40 nM for 48 h, a leftward shift of the caffeine cumulative concentration-response curve was observed (Figure 1A, *n* = 6). When the inhibitory concentration of 50% (IC_50_) was analyzed, a significantly lower concentration (0.30 ± 0.51 nM) was required to achieve 50% caffeine-induced relaxation when tracheal rings were exposed to TES compared with the control group (0.39 ± 0.04 nM) (Figure 1B, left panel, *n* = 6). The addition of tetraethylammonium (TEA 1 mM), a non-selective K^+^ channel blocker, shifted the curve to the right (Figure 1A, *n* = 6), but no statistical differences in IC_50_ were observed compared with the control group. Similarly, no differences were found in maximal relaxation among the three groups (Figure 1B, right panel). Because the TES-induced enhancement of relaxation is abolished in the presence of TEA, these results suggest that K^+^ channels are involved in the androgen action.

Since caffeine exerts a variety of functions in airway smooth muscle (ASM), including the opening of ryanodine receptors and the consequent release of Ca^2+^ from the SR [63], we decided to test another methylxanthine known for its relaxing effects via inhibition of phosphodiesterases (PDEs). The non-selective PDE inhibitor isobutylmethylxanthine (IBMX) [64,65] relaxed tracheal rings precontracted with histamine 10 μM in a concentration-dependent manner. As in the caffeine experiments, tissues preincubated with 40 nM TES for 48 h showed a significant leftward shift of the cumulative IBMX concentration-response curve, whereas TEA 1 mM shifted this curve to the right (Figure 2A, *n* = 6). The IBMX IC_50_ value was significantly lower than the IC_50_ of the sham group when TES was tested (3.22 ± 0.48 μM vs. 2.24 ± 0.38 μM, respectively, *p* < 0.05). In addition, the TEA group showed a significantly higher IC_50_ (4.31 ± 0.57 μM, *p* < 0.01) than the control group (3.22 ± 0.48 μM) (Figure 2B, right panel). Neither TES nor TES + TEA differed from the control group’s maximal response to IBMX (Figure 2B, left panel). These results strongly suggest that the TES-enhanced tracheal smooth muscle relaxation by methylxanthines is related to the activity of K^+^ channels, and because of the long exposure time to the androgen, we may be observing a genomic effect.

### 2.2. Theophylline-Induced Relaxation of Tracheal Smooth Muscle Is also Enhanced by Testosterone, and K^+^ Channels Are Involved in This Androgen’s Action

In view of the previous results, we decided to test whether testosterone was capable of improving ASM relaxation over the clinically used methylxanthine, theophylline. This compound is one of the most commonly prescribed drugs for the treatment of asthma and is recommended as second-and third-line therapy in patients with poor control of this ailment [2,4]. Like other methylxanthines, this drug is a non-selective PDE inhibitor that relieves bronchospasm [66]. Cumulative concentration-response curve to theophylline relaxed tracheal rings precontracted with histamine (10 μM). After 48 h of tissue preincubation with TES 40 nM, the concentration-response curve to theophylline was shifted to the left (Figure 3A, *n* = 8). The potentiating effect of TES was abolished by the acute addition of TEA (1 mM). Figure 3B (right panel) shows that TES decreased the IC_50_ of theophylline compared with the control group (78.59 ± 8.58 μM vs. 105.76 ± 06.03 μM, respectively, *p* < 0.01). Conversely, TEA significantly increased theophylline’s IC_50_, confirming that TES-induced upregulation of K^+^ channels is probably involved in the enhanced response to methylxanthines. Maximal relaxation was not altered by treatment with TES or TES+TEA (Figure 3B, left panel). Importantly, plasmatic therapeutic concentrations of theophylline lay in the range of 10–20 mg/L (55–110 μM) [66,67]. Note that the IC_50_ values for the control (105.76 ± 06.03 μM) and TES (78.59 ± 8.58 μM) groups are within this range.

### 2.3. Testosterone Increases Theophylline-Induced K^+^ Currents via the Androgen Receptor Signaling Pathway in Tracheal Myocytes

In patch-clamp experiments, K^+^ currents (IK^+^) were elicited by a series of depolarizing pulses (−60 to 50 mV in 10 mV steps) in single tracheal smooth muscle cells. This procedure generated outward IK^+^, which peaked at 0.5 ± 0.07 nanoamperes (nA) at the maximum voltage tested (Figure 4A, closed circle). The addition of increasing concentrations of theophylline (10, 32, 100, 320 µM) enhanced the increment in IK^+^ in a concentration-dependent manner (Figure 4A). The highest concentration of theophylline used (320 µM) resulted in IK^+^ reaching a peak of 2.4 ± 0.12 nA at 50 mV (Figure 4A). Myocytes incubated with TES 40 nM for 48 h showed even higher IK^+^ depending on the theophylline concentration used (Figure 4B). However, adding the androgen receptor antagonist Flu 3.2 μM abolished the increase in IK^+^ induced by TES incubation, suggesting that the effect of the androgen is through a genomic pathway (Figure 4C). For illustration and appropriate analysis of the effects of TES and Flu, the increase in IK^+^ was compared regarding the concentration of theophylline used. Figure 1D shows that IK^+^ induced by the protocol of depolarizing pulses alone increased significantly from −10 mV ahead when myocytes were exposed to TES; this effect was reversed by flutamide. The addition of theophylline at different concentrations (10, 32, 100, 320 µM) increased the IK^+^ over the depolarizing steps protocol (Figure 4E–H). Moreover, the increased IK^+^ was further significantly increased by TES from −30 mV (Figure 4E), −10 mV (Figure 4F), or −50 mV ahead (Figure 4G,H), and this effect was also reversed by flutamide. Overall, these results confirm the genomic nature of the effect of TES on theophylline-induced IK^+^.

### 2.4. The Effect of Testosterone Is Mediated Mainly by the Upregulation of K_V_ Channels

To elucidate which type of K^+^ channel is involved in the TES effect, we performed patch-clamp experiments with a concentration of theophylline that approached 100% relaxation in the presence of TES and represented the optimal K^+^ current response. In control cells, the increase in IK^+^ induced by theophylline 320 µM was partially blocked by 4-aminopyridine (4-AP, 3 mM), a blocker of the delayed rectifier K^+^ channel (K_V_) from −40 mV ahead. Further addition of iberiotoxin (IBTX, 100 nM), the specific blocker of high conductance Ca^2+^-activated K^+^ channels (BK_Ca_), abolished the remaining theophylline-induced IK^+^. The area under the curve (AUC) analysis revealed that the contribution of K_V_ and BK_Ca_ channels to the theophylline response was 72.17% and 27.83%, respectively (Figure 5A). However, when airway myocytes were incubated with the androgen, the increase in theophylline-induced IK^+^ was significantly higher, as shown in Figure 5. The resulting IK^+^ was almost attenuated by 4-AP 3 mM from −50 mV forward, and the remanent current (from 20 mV ahead) was abolished by IBTX 100 nM. Calculated AUC showed that the contribution of K_V_ and BK_Ca_ channels to the theophylline response in the presence of TES was 82.37% and 17.62%, respectively (Figure 5B). To make an integral estimate of the relative contribution of Kv and BK_Ca_ channels to the K^+^ current induced by theophylline and enhanced by TES, we run patch-clamp experiments by administering 4-AP and IBTX in the opposite order. Under control conditions, BK_Ca_ channels were responsible for 35.17% of the theophylline response, whereas K_V_ channels accounted for 64.83% (Figure 6A). When myocytes were exposed to TES 40 nM, a greater contribution of K_V_ channels (68.2%) was observed, while BK_Ca_ channels were responsible for only 31.8% of the theophylline-induced K^+^ currents (Figure 6B). Taking the two administration sequences of 4-AP and IBTX together, under control conditions, K_V_ channels are involved in 68.5% of theophylline-evoked K^+^ current, and BK_Ca_ channels correspond to 31.5%. Exposure of airway myocytes to TES increased the contribution of K_V_ channels to 75.29%, and BKCa was reduced to 24.71%. As can be seen, in both cases, K_V_ channels play a predominant role in the theophyllne response. More importantly, the AUC for K_V_ channels increased when cells were exposed to TES, by 10.2% and 3.37%, respectively, whereas the AUC for BK_Ca_ channels decreased. Overall, these results suggest that the androgen effect on theophylline-induced IK^+^ is mainly due to the upregulation of the delayed rectifier K^+^ channel.

### 2.5. K_V_1.2, K_V_1.5, and BK_Ca_ Subtypes Are Expressed in Guinea Pig Airway Smooth Muscle and Testosterone Only Increases the Expression of K_V_ Channels

We found that K_V_1.2, K_V_1.5, and BK_Ca_ (K_Ca_1.1) channels are present in guinea pig ASM, as reported in the literature [28,33,34]. Moreover, immunofluorescence studies showed that incubation of TES 40 nM for 48 h increased the expression of K_V_1.2 (Figure 7) and K_V_1.5 (Figure 8). These results were corroborated by the mean fluorescence, which is observed in Figure 7D and Figure 8D. However, ASM chronic exposure to TES failed to increase BK_Ca_ expression (Figure 9). No fluorescent signal was detected in the negative controls performed by incubation of the respective blocking peptide or without incubation of the primary antibody, confirming the specificity of the antibodies against K_V_1.2, K_V_1.5, and K_Ca_1.1. The colocalization of the three K^+^ channel subtypes in ASM was confirmed by detecting smooth muscle α-actin (Figure 7A,B, Figure 8A,B, and Figure 9A,B, last panel).

## 3. Discussion

According to the present study, chronic exposure to TES increases the protein levels and functionality of K_V_1.2 and K_V_1.5 channels in guinea pig ASM and enhances theophylline-induced relaxation via a genomic pathway. Because TEA reversed this effect, the results of the organ bath experiments demonstrate the importance of K^+^ channels in the theophylline relaxation response. Theophylline-induced whole-cell K^+^ currents in airway myocytes were improved by TES, and this enhancement was abolished by flutamide. Moreover, 4-AP nearly abolished the effect of TES on IK^+^ elicited by theophylline, emphasizing the importance of K_V_ channels for the androgen´s action.

The use of bronchodilators is essential for symptomatic control of asthma, as these medications directly relax ASM. Currently, there are three main types of bronchodilators that can be used alone or in combination: β_2_-AR agonists, muscarinic receptor antagonists, and methylxanthines [68]. Theophylline, among the available methylxanthines, is still one of the most widely used medications for treating asthma since it is affordable and easily accessible, despite the development of certain substituted derivatives [69,70,71]. The widely used dietary methylxanthines caffeine and theobromine share structural similarities and bioactivity with this drug [72]. In the current study, we found that caffeine, IBMX, and theophylline relax guinea pig ASM precontracted with histamine in a concentration-dependent manner. Furthermore, chronic exposure (48 h) of tracheal tissues to a physiological concentration of TES (40 nM) enhanced the relaxation produced by all three methylxanthines tested (Figure 1, Figure 2 and Figure 3). The bronchodilator action of theophylline in vitro in human airways has been investigated. In 1984, Guillot and colleagues reported that theophylline relaxes human bronchial rings precontracted with acetylcholine, showing an IC_50_ of 150 µM [73]. Additionally, Miura and colleagues reported that this methylxanthine relaxes the spontaneous tone of human ASM with an IC_50_ of 32 µM [39]. According to our findings, 105.76 ± 6.03 µM of theophylline is required to reduce the histamine-induced contraction in guinea pig tracheal rings by 50%. This theophylline concentration is at the upper limit of the maximum advised for therapeutic purposes (110 μM). Nevertheless, theophylline’s IC_50_ was significantly lowered by TES to 78.59 ± 8.58 μM (Figure 3B). In addition, we observed that 100 µM of theophylline, a concentration reaching the maximum recommended, relaxed control tissues by ~45%, whereas TES-treated tissues were relaxed by ~60%. These findings suggest that theophylline might be utilized at safer doses and that male asthma patients may respond better to this medication.

One of the most important mechanisms by which methylxanthines cause ASM relaxation is the inhibition of PDEs [74]. From the approximately 21 genes encoding 11 subfamilies of PDEs, over 200 gene products are formed by alternative splicing. The primary isoforms responsible for cAMP degradation in ASM are members of the PDE4 family, particularly PDE4D splice variants, with a minimal contribution from PDE4B and PDE3 [20,21,74]. Indeed, PDE4D has been shown to be the most abundant PDE4 isoform in mouse ASM, where it regulates cholinergic contraction [75]. In this regard, the response to carbachol and methacholine is greatly reduced and abolished, respectively, in PDE4D null mice [75,76]. In addition, PDE isoenzymes are essential regulators in the control of cAMP signaling [74]. The increase in cAMP represents the main relaxant signaling pathway in ASM [77]. To induce relaxation, cAMP activates PKA, which phosphorylates several target proteins including K^+^ channels [26,28,32,78]. These channels regulate ASM tone and mediate relaxation by causing membrane hyperpolarization and reducing membrane excitability [33,36,39,79,80,81]. Among the various K^+^ channels expressed in the ASM, Ca^2+^-activated K^+^ (K_Ca_) channels and K_V_ channels are the most important types [33,34,38]. According to their electrophysiological properties, K_Ca_ channels are divided into three subfamilies: high conductance (BK_Ca_, MaxiK, K_Ca_1.1), intermediate conductance (IK_Ca_, K_Ca_3.1), and low conductance (SK_Ca_, K_Ca_2.1, 2.2, and 2.3) [28,82,83]. K_Ca_ are primarily activated by an increase in intracellular Ca^2+^ concentration ([Ca^2+^]_i_) and through the cAMP-PKA signaling cascade [28,29,78]. BK_Ca_ is the most significant subtype of K^+^ channels that induce relaxation of ASM when the tissue is stimulated with ATP, serotonin, or β_2_-adrenergic agonists such as salbutamol [36,37,38,78]. On the other hand, K_V_ channels are activated mainly by changes in membrane voltage and by cAMP-PKA signaling [32,33,34]. Since their blocking with 4-AP improves smooth muscle tension, these channels are the most significant K^+^ subtypes in maintaining ASM tone [33]. In ASM, the most relevant subtypes characterized are K_V_1.2, K_V_1.5, and K_V_7.5 [32,33].

One of the first insights into the role of K^+^ channels in the theophylline relaxation response was obtained by Miura and colleagues [39]. They found that charybdotoxin, a non-specific blocker of K_Ca_ channels [84], significantly shifted the theophylline response curves to the right in human bronchial rings [39]. Moreover, we have recently shown that TES enhances K_V_ and BK_Ca_-mediated currents when tracheal myocytes are stimulated with salbutamol [37]. Therefore, in the present work, we sought to investigate whether TES may upregulate K^+^ channels and enhance the theophylline relaxation response. Furthermore, since this drug leads to an increase in intracellular cAMP levels, it is conceivable that K^+^ channels are involved in the enhanced response to theophylline. In this regard, we found that TEA, the non-selective blocker of K^+^ channels, annulled the potentiation (induced by chronic exposure to TES) of the ASM relaxation elicited by caffeine, IBMX, and theophylline (Figure 1, Figure 2 and Figure 3). In addition, the use of TEA significantly increased the IC_50_ of IBMX and theophylline (Figure 2B and Figure 3B), indicating the importance of K^+^ channels in the response to methylxanthines and suggesting that the effect of TES on K^+^ channels in ASM is what causes the enhancement of methylxanthine-mediated relaxation. The usual concentrations of TEA, which block both K_Ca_ and K_V_ channels in this tissue, range from 1 to 30 mM [82,85,86]. We chose 1 mM to achieve the blockade of K^+^ channels, a concentration that allowed us to observe the relaxation of ASM triggered by methylxanthines in organ bath experiments.

Several works have shown that TES exerts genomic and nongenomic effects on K^+^ channels in smooth muscle. For instance, nongenomic effects were seen in cultured human corporal smooth muscle cells stimulated with 200 nM of TES (for 40 min) and subjected to patch-clamp assays under the cell-attached configuration, which showed increases in the single-channel activity of BK_Ca_ channels [61]. The same TES concentration augmented whole-cell currents mediated by ATP-sensitive K^+^ (K_ATP_) channels. These TES effects were abolished by TEA 1 mM (a non-specific blocker of K_Ca_ channels) and glibenclamide 10 µM (a blocker of K_ATP_ channels) [61]. Interestingly, the same work reported that the incision of cell-attached patches into an inside-out configuration abrogated the effect of TES, highlighting the involvement of a signaling cascade and a second messenger induced by TES in these cells [61]. Meanwhile, in the human umbilical artery, TES (1–100 µM) induced relaxation when the tissue was precontracted with histamine (His), serotonin (5-HT), or KCl [87]. The role of K_V_ and K_Ca_ channels was demonstrated as 4-AP reduced the TES-associated effect in 5-HT and His-contracted arteries, whereas TEA diminished the relaxing effect of TES on 5-HT and KCl-stimulated arteries. Furthermore, flutamide did not alter TES-induced relaxation in His, 5-HT, or KCl-contracted arteries, ruling out the possibility of a genomic pathway [87]. In porcine coronary arteries, the highly selective blocker of BK_Ca_ channels, iberiotoxin (IBTX), decreased the relaxation induced by TES (25 µM) when the tissues are contracted with prostaglandin F2α (PGF_2α_) [60]. In addition, 200 nM TES during 40 min increases the activity of single BK_Ca_ channels in porcine coronary artery myocytes, and this effect is mimicked by the cell-permeable cGMP analog, 8-bromo-cGMP. Moreover, enzyme immunoassays showed that micromolar concentrations of TES cause an accumulation of cGMP in coronary arteries, suggesting that this second messenger may mediate the effects of TES on BK_Ca_ channels [60]. Regarding the genomic effects of TES on K^+^ channels, Masuda and colleagues showed that QT intervals in the ECG of castrated rats significantly decreased by intraperitoneally administered TES (1000 µg/kg/day for 14 days) [62]. This effect is probably mediated by upregulation of K_V_7.1, as this protein is augmented after this maneuver. Moreover, 24 h but not 5 min administration of the 5α-reduced metabolite of TES, DHT (3 nM), improves K_V_ currents in cardiomyocytes, strongly suggesting that the effect of the androgen on shortened QT intervals may be via a genomic pathway involving upregulation of Kv7.1 [62].

To further investigate the role of K^+^ channels in the TES effect on methylxanthine-induced ASM relaxation, we performed patch-clamp experiments. Most of the above-discussed studies on the effects of TES on K^+^ channels were performed with non-physiological concentrations of this androgen. Our results showed that chronic exposure of tracheal myocytes to a physiological concentration of TES (40 nM) resulted in an increase in IK^+^ triggered by depolarizing pulses (Figure 4D) and by several concentrations of theophylline below and above therapeutic levels (Figure 4E–H). Both effects on IK^+^ were abolished by flutamide (Figure 4C), indicating that the theophylline-induced enhancement of IK^+^ after TES incubation was due to a genomic effect. Another set of experiments showed that K_V_ channels and BK_Ca_ channels are the major K^+^ channel subtypes in theophylline-induced IK^+^ in ASM cells. In this context, the IK^+^ produced by stimulation of tracheal myocytes with 320 µM theophylline (a concentration that causes more than 80% relaxation in control and TES-treated tissues, Figure 3) is determined to be ~69% by the activity of K_V_ and ~31% by BK_Ca_ channels. When myocytes were exposed to TES, 4-AP blocked theophylline-induced IK^+^ by ~75%, whereas the use of IBTX, the specific BK_Ca_ blocker, reduced IK^+^ by ~25%. These results indicate a predominant role of K_V_ channels in theophylline response and suggest that TES upregulates these proteins in ASM, while no effect on BK_Ca_ channels is observed. Although specific K_V_ channel blockers have been developed, 4-AP represents one of the most reliable pharmacological tools to characterize these proteins. In addition, evidence suggests that the major subtypes in ASM are K_V_1.2, K_V_1.5, and K_V_7.5. K_V_1.2 and K_V_.15 are blocked by 4-AP, whereas Kv7.5 channels are not, ruling out their involvement in the theophylline response. BK_Ca_ as well as K_V_1.2 and K_V_1.5 channels can be activated by cAMP-PKA signaling [28,30,31,88]; thus, it is conceivable that theophylline induces their opening. Because TES causes genomic actions on K^+^ channels and in the theophylline relaxation response and considering the most important subtypes of K^+^ channels in ASM, we investigated whether upregulation of these proteins was involved. In this context, we found by immunofluorescence that K_V_1.2 and K_V_1.5 were increased in the ASM of guinea pigs chronically exposed to TES (Figure 7 and Figure 8). Despite BK_Ca_ channels contributing to theophylline-induced IK^+^, we did not observe any increase in fluorescence intensity for this protein (Figure 9). In addition, further research is needed to investigate whether this androgen exerts a nongenomic effect at physiological concentrations on the IK^+^ in ASM.

The importance of plasmatic TES concentrations in diminishing the severity of asthma symptoms when young asthmatic males enter puberty is well documented. In contrast, symptoms worsen in girls of the same age [43,46]. In fact, compared to men, women have a higher risk of developing asthma during their lifetime and a more severe form of the disease [89]. Therefore, decreased asthmatic condition is likely associated with greater TES plasmatic concentrations (6–50 nM) in teenage boys [45,46,90]. This propensity predominates throughout a man’s lifespan and declines with age. Research about TES genomic effects (at nanomolar concentrations) on asthma symptoms primarily focuses on airway inflammation since this androgen adversely affects type 2 inflammation that is caused by CD4+ Th2 cells (Th2) and group 2 innate lymphoid cells (ILC2s) [58,59]. These lymphocytes contribute to allergic airway inflammation in asthmatic patients, which is mediated by mast cells, macrophages, basophils, eosinophils, and enhanced immunoglobulin E (IgE) levels [40,91]. Additionally, some patients who do not develop type 2 airway inflammation exhibit IL-17A-mediated neutrophil inflammation [89] which is also reduced by nanomolar concentrations of TES [40,43,92]. In this regard, theophylline is a well-known anti-inflammatory drug with several proposed action mechanisms [2,3,93]. One of these mechanisms is the inhibition of PDEs located in inflammatory cells, which leads to increased cAMP concentration. This second messenger prevents the release of inflammatory mediators and cytokines [94,95], as well as ASM cell proliferation [20], which is crucial in the pathophysiology of asthma [96,97]. Whether TES exerts a genomic effect on PDEs in the lung to modulate inflammation is not known and should be investigated. In addition, the regulation of PDE4 variants in ASM needs further investigation. In this context, TES might downregulate PDEs in this tissue to promote an enhanced relaxation response to theophylline. However, our results clearly demonstrate that the pharmacological blockade of K^+^ channels is sufficient to abolish the androgen-mediated genomic effect. Finally, gender differences in asthma therapy have not been extensively studied [89]. In this regard, it has been suggested that women may benefit more from anti-leukotriene therapy than men [89,98]. In the present work, we hypothesize that male teenagers and men are likely to respond better to theophylline-based bronchodilator therapy than girls and women.

## 4. Materials and Methods

### 4.1. Experimental Animals

Male Hartley guinea pigs (300–450 g) bred under conventional conditions in our institutional animal facilities (filtered conditioned air at 21 ± 1 °C, 50–70% humidity, and sterilized bed) and fed with Harlan^®^ pellets and sterilized water were used. The protocol was approved by the Scientific and Bioethical Committees of the Facultad de Medicina, Universidad Nacional Autónoma de México. Experiments were performed in accordance with published guidelines for the care and use of animals approved by the American Physiological Society (https://www.physiology.org/career/policy-advocacy/policy-statements/care-and-use-of-vertebrate-animals-in-research?SSO=Y, 2014, accessed on 22 November 2022) and the National Institutes of Health Guide for the Care and Use of Laboratory Animals (NIH, Eight Edition, National Academies Press, 2011, ISBN-13: 978-0-309-15400-0) as well as Mexican National Animal Welfare Laws, and Protection and the General Health Law for Health Research (NOM-062-Z00-1999).

### 4.2. Organ Baths

Guinea pigs were submitted to euthanasia with Pentobarbital sodium (35 mg/kg, i.p.) and exsanguinated. The trachea was removed, carefully dissected, and freed from connective tissue. Testosterone exposure was performed as previously described [37]. Briefly, eight rings, each approximately 4 mm long, were obtained from a single guinea pig trachea. Each ring was placed in an Eppendorf tube filled with 1 mL of Krebs solution containing (in mM): 118 NaCl, 25 NaHCO_3_, 4.6 KCl, 1.2 KH_2_PO_4_, 1.2 MgSO_4_, 11 glucose, and 2 CaCl_2_. Krebs solution was previously saturated with oxygen for 30 min by bubbling 5% CO_2_ in oxygen, maintaining the pH at 7.4. Testosterone 40 nM was added to some tracheal preparations, and all tubes were sealed with parafilm and stored at 9 °C for 48 h. Cooling of the tissues reduced metabolism and allowed their preservation. In this context, it was shown in Chinese hamster ovary cells that cold shock with ice-cold phosphate buffer saline (PBS, about 0–4 °C) reduced protein synthesis by 84%. This loss of protein synthesis capacity can be recovered by a short period of 20 min at 28 °C, which allows energy metabolism. Moreover, this procedure does not harm the translational machinery [99]. Thus, our tissues should have a protein synthesis rate of about 16%. After 48 h, each ring was suspended in a 5 mL organ bath filled with Krebs solution and maintained at 37 °C with 5% CO_2_ bubbled in oxygen at pH 7.4. The tracheal rings were attached with a silk thread to an isometric force transducer (model FT03; Grass Instruments, West Warwick, RI, USA) connected to a signal conditioner (CyberAmp 380, Axon Instruments, Foster City, CA, USA) and an analog-to-digital interface (Digidata 1440A; Axon, USA). Data were recorded and analyzed using acquisition and analysis software (AxoScope version 10.2; Axon). At the beginning of the experiments, all tissues were subjected to a resting tension of 1 g for 60 min. Tracheal rings were then stimulated three times with KCl 60 mM to acclimate the tissues and optimize the contractile apparatus. To evaluate the relaxing response to three different methylxanthines, all tracheal rings were precontracted with 10 μM histamine, and once maximal contraction was achieved, cumulative concentrations of caffeine (0.2, 0.4, 0.6, 0.8, 1, 1.2, 1.4, 1.6, 1.8, and 2 mM), isobutylmethylxanthine (IBMX, 0.32, 0.56, 1, 1.7, 3.2, 5.6, 10, 17, 32 and 56 μM), or theophylline (Theo, 1, 3.2, 10, 32, 100, 320, 1000, 32,000, and 10,000 μM) were added. To assess the role of K^+^ channels in methylxanthines-induced relaxation, in another set of experiments the non-selective K^+^ channel blocker tetraethylammonium (TEA 1 mM) was added for 10 min before the cumulative concentration-response curve with caffeine, IBMX, or theophylline.

### 4.3. Patch-Clamp Studies

Tracheal myocytes were obtained and cultured as follows. Guinea pig tracheal smooth muscle was freed from connective tissue and epithelium and placed in 5 mL of Hanks’ balanced salt solution containing 2 mg L-cysteine and 0.04 U/mL papain. The pH was adjusted to 7.4 with 1 M NaHCO_3_, and the tissues were incubated at 37 °C for 10 min. Subsequently, the preparations were washed with Leibovitz’s (L-15) medium to remove the enzyme excess, and the tracheal tissue was then placed in Hanks’ balanced salt solution containing 1 mg/mL collagenase type I for 10 min at 37 °C. The myocytes were gradually dispersed by mechanical shaking of the tissue until the detached cells could be observed under the light microscope. To stop enzymatic activity, L-15 medium was used again, the cells were centrifuged at 600 rpm for 5 min at 20 °C, and the supernatant was discarded. This last procedure was repeated once. For myocyte culture, the cell pellet was resuspended in minimal essential medium containing 5% fetal bovine serum, 2 mM L-glutamine, 10 U/mL penicillin, 10 μg/mL streptomycin, and 15 mM glucose and plated on rounded coverslips previously coated with sterile rat tail collagen. To some wells, TES 40 nM or flutamide 3.2 μM (an androgen receptor antagonist, added 30 min before the addition of the androgen), was administered, and then, cells were grown for 48 h at 37 °C in 5% CO_2_ in oxygen. Myocytes spread in the coverslip were then placed at the bottom of the 0.7-mL perfusion chamber and cells were allowed to settle down. The chamber was perfused by gravity with an external solution containing the following (in mM): 130 NaCl, 1 CaCl_2_, 0.5 MgCl_2_, 3 NaHCO_3_, 1.2 KH_2_PO_4_, 5 KCl, 0.1 niflumic acid, 10 glucose, and 10 HEPES (pH 7.4, adjusted with NaOH). All experiments were conducted at room temperature (~22 °C).

Membrane K^+^ currents (IK^+^) activated by depolarizing voltage steps (Voltage Clamp approach) were recorded using the standard whole-cell configuration and an Axopatch 200A amplifier (Axon Instruments). Patch pipettes were made of 1B200F-6 glass (Word Precision Instruments, Sarasota, FL, USA) with a micropipette puller (P-87, Sutter Instruments Co., Novato, CA, USA). Each patch pipette had a resistance between 2 and 4 MΩ and was filled with an internal solution containing the following (in mM): 5 NaCl, 140 K^+^ gluconate, 5 ATP, 0.1 GTP, 5 HEPES, 1 EGTA, and 0.1 leupeptin. The pH was always adjusted to 7.3 with KOH. The generated currents were filtered at 1–5 KHz, digitized at 10 KHz with the Digidata 1440A digitizer (Axon Instruments), and finally analyzed using the software pClamp v10.2 (Axon Instruments).

Single tracheal myocytes were subjected to a series of depolarizing square-wave pulses with potentials ranging from −60 to +50 mV in 10 mV steps from a holding potential of −60 mV for 500 ms at 1 Hz. This procedure allowed us to examine outward K^+^ currents. After the control protocol and recording of basal K^+^ currents, a concentration-response curve to theophylline (10, 32, 100, and 320 μM) was generated for each experimental group. To evaluate the role of delayed rectifier K^+^ channels (K_V_) and high conductance Ca^2+^-activated K^+^ channels (K_Ca_1.1, BK_Ca_), tracheal myocytes were subjected to the same pulse protocol described above and perfused with theophylline 320 μM. After recording the resulting current, K_V_ channels were blocked with 4-aminopyridine (4-AP, 3 mM), and BK_Ca_ channels were blocked with iberiotoxin (IBTX, 100 nM). The changes in currents from the protocols explained above were assessed as the maximum current peak at each voltage tested.

### 4.4. Double Immunofluorescence

Tracheas from guinea pigs were obtained as previously described and then incubated with or without TES 40 nM for 48 h at 9 °C [37]. Tracheal tissues were then fixed in 4% paraformaldehyde in phosphate-buffered saline (PBS) for 4 h after gradual dehydration in 30% sucrose/PBS for 17 h before being included in O.C.T. compound (Tissue Tek, CA, USA) and frozen at −60 °C. Frozen sections (10 μm thick) were obtained with a cryostat (Ecoshel ECO-1900, Pharr, TX, USA) and mounted on glass slides. Afterwards, slides were transferred to PBS and permeabilized with 0.001% Tween 20 in TBS. To block non-specific binding to proteins, cryosections were treated with 10% horse serum for 2 h at room temperature. The primary antibodies anti-K_V_1.2 (GeneTex, GTX16774, Irvine, CA, USA), anti-K_V_1.5 (GeneTex, GTX16716), and the antibody against Maxi K^+^ channel alpha (K_Ca_1.1, GeneTex, GTX54874) were incubated at a dilution of 1:30 overnight at 4 °C. The secondary antibody Alexa488 donkey anti-rabbit IgG (Life Technologies, Foster City, CA, USA) was incubated for 20 min (1:500). Anti-α-actin (sc-58669, Santa Cruz Biotechnology, Dallas, TX, USA), diluted 1:600, was incubated overnight as the next primary antibody. The secondary antibody Alexa Fluor 555 donkey anti-mouse (Life Technologies) at a dilution of 1:600 was applied. The nuclei in all tissue slices were counterstained with 4’,6-diamidino-2-phenylindole (DAPI, Life Technologies). Slides were maintained with Fluoromount aqueous mounting medium (Merck, Darmstadt, Germany). A blocking peptide was used for K_V_1.2, and negative controls without primary antibodies (K_V_1.5 and K_Ca_1.1) had no fluorescent signal. Immunofluorescence was observed using an LSM 880 Zeiss confocal microscope (Oberkochen, Germany). For display purposes, images were merged in which K_V_1.2, K_V_1.5, and K_Ca_1.1 are green, α-actin is red, and nuclei are blue.

Fluorescent quantification: All immunofluorescences were undertaken in parallel, the parameters of pinhole size, detector, amplifier gain, amplifier compensation, and laser intensity were first set for each anti-K_V_1.2, anti-K_V_1.5, and K_Ca_1.1 channels, using negative controls, and this setting was used for all experiments. Non-overlapping images for each channel were acquired and stored as 12 to 16-bit fluorescent TIFF images. For analysis, only the image of each anti-K_V_1.2, anti-K_V_1.5, and K_Ca_1.1 channel was used. Then, these were imported into ImageJ software 1.5.3 to determine immunofluorescent intensity with ROI analysis. Mean fluorescence intensity (counts/pixel) in the smooth muscle region was determined by selecting a random representative area of 3138 µm^2^ per sample. A minimum of 9 areas of 3 histological sections per sample were analyzed from 4 to 6 individuals.

### 4.5. Drugs and Chemicals

Theophylline (1,3-dimethylxanthine), caffeine (1,3,7-trimethylxanthine), isobutylmethylxanthine (3-isobutyl-1-methylxanthine, IBMX), histamine, testosterone (4-androstene-17β-ol-3-one, TES), and flutamide were obtained from Sigma Chem. Co. (St. Louis, MO, USA). Notably, 4-Aminopyridine (4-AP) was purchased from Research Chemical LTD (Word Hill, MA, USA), and iberiotoxin was acquired from Santa Cruz Biotechnology (Dallas, TX, USA). TES and flutamide were diluted in absolute ethanol and IBMX in DMSO. The highest percentage used was 0.1% *v*/*v* of the vehicle.

### 4.6. Statistical Analysis

In organ bath experiments, tracheal smooth muscle relaxation induced by caffeine, IBMX, or theophylline was evaluated by the inhibitory concentration of 50% (IC_50_) and maximum relaxation. Each cumulative concentration-response curve for the three different methylxanthines was used to calculate the IC_50_, which was computed by straight-line regression as -Log [M] using ED50 plus v1.0 software. These data were analyzed by repeated-measures analysis of variance, followed by Dunnett’s multiple comparison test. K^+^ currents in individual cells at each voltage step were analyzed by one-way analysis of variance followed by Dunnett’s multiple comparison tests for unpaired data and by repeated-measures analysis of variance followed by Student–Newman–Keuls’ test for paired data. The AUC of theophylline-induced K^+^ currents was calculated using SigmaPlot 12.0 software and plotted as a percentage of the area between control and theophylline traces. Differences in mean immunofluorescence were analyzed with an unpaired two-tailed *t*-test. Each “*n*” value in every experiment represents a distinct animal. Data and figures presented in the manuscript are expressed as mean ± S.E.M. Statistical significance was set at *p* < 0.05 bimarginally.

## 5. Conclusions

Our results indicate that chronic ASM exposure to a physiological TES concentration (in the nanomolar range) promotes the upregulation of K_V_1.2 and K_V_1.5 and consequently favors the relaxation response to theophylline. Therefore, gender should be considered when prescribing this drug to asthma patients, as teenage boys and men are likely to respond better than females.

## Figures and Tables

**Figure 1 ijms-24-05884-f001:**
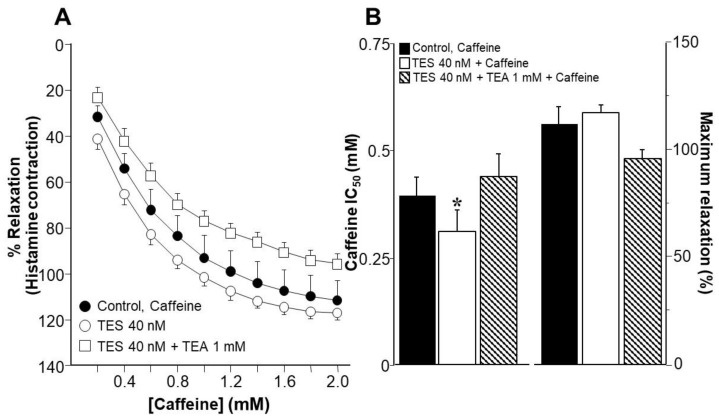
Chronic incubation with testosterone (TES) potentiates caffeine-induced guinea pig tracheal smooth muscle relaxation, whereas acute (10 min) administration of tetraethylammonium (TEA) suppresses this androgen’s action. (**A**) Cumulative caffeine concentration-response curve showing relaxation of tracheal rings precontracted with histamine (10 μM). Tissue incubation with TES (40 nM, *n* = 6) for 48 h induced a significant leftward shift of the concentration-response curve to caffeine. In contrast, administration of the non-selective K^+^ channel blocker TEA (1 mM) 10 min before stimulation with caffeine shifted the curve to the right. (**B**) The bar graphs summarize that TES 40 nM significantly reduced the caffeine inhibitory concentration by 50% (IC_50_, right panel) compared with the control group. The maximum response (left panel) to caffeine did not differ when TES or TES + TEA were tested. Symbols and bars represent mean ± S.E.M., * *p* < 0.05. Repeated measure analyses of variance were performed, followed by Dunnett’s multiple comparison tests.

**Figure 2 ijms-24-05884-f002:**
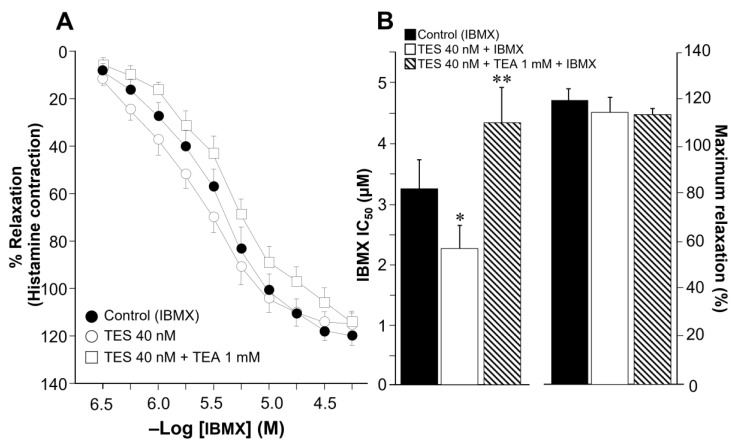
Isobutylmethylxanthine (IBMX) induces guinea pig tracheal smooth muscle relaxation, which is enhanced by chronic exposure to testosterone (TES), while acute treatment with tetraethylammonium (TEA) abolishes the enhancement, suggesting that K^+^ channels are involved in TES action. (**A**) Cumulative concentration curve of IBMX relaxes tracheal rings precontracted with histamine (10 μM). After 48 h of preincubation of tracheal rings with TES (40 nM, *n* = 6), the concentration-response curve to IBMX was significantly shifted to the left. This TES effect was abolished by acute addition (10 min) of 1 mM TEA, a non-selective K^+^ channel blocker. (**B**) The bar graph shows that TES decreases the inhibitory concentration by 50% (IC_50_) of IBMX compared with the control group, while TEA 1 mM increases this parameter, indicating the involvement of K^+^ channels in IBMX relaxation and testosterone effects. Maximal relaxation was not altered by TES treatment. Symbols and bars represent mean ± S.E.M., * *p* < 0.05, ** *p* < 0.01. Repeated measure analyses of variance were performed, followed by Dunnett’s multiple comparison tests.

**Figure 3 ijms-24-05884-f003:**
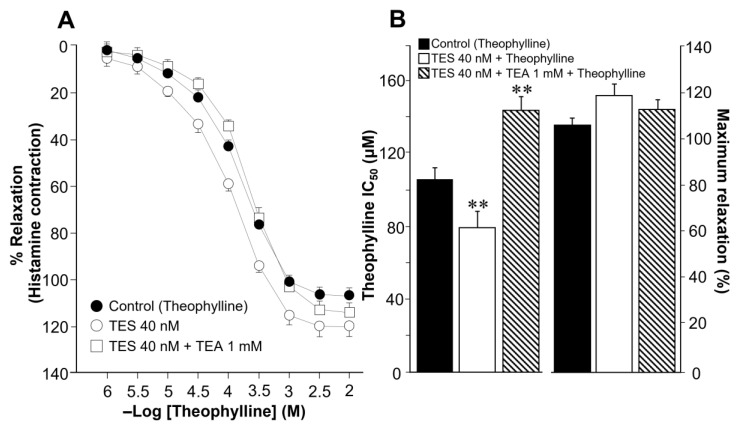
Testosterone (TES) incubated for 48 h enhances theophylline-induced relaxation of guinea pig tracheal rings, whereas tetraethylammonium (TEA) blocks TES potentiation, indicating that K^+^ channels are involved in the androgen effect. (**A**) Cumulative concentration curve of theophylline relaxes tracheal rings precontracted with histamine (10 μM). Incubation of the tissues with TES 40 nM (*n* = 8) for 48 h results in a leftward shift of the concentration-response curve to theophylline. When TEA is applied, the enhancement of the theophylline concentration-response curve caused by TES is abolished. (**B**) Bar graphs show that TES 40 nM significantly decreases the theophylline inhibitory concentration by 50% (IC_50_, right panel) compared with the control group. The addition of TEA 1 mM increases the theophylline IC_50_, suggesting the role of K^+^ channels in the androgen action. The maximal response (left panel) to theophylline was not different between groups. Symbols and bars represent mean ± S. E. M., ** *p* < 0.01. Repeated-measured analyses of variance were performed, followed by Dunnett’s multiple comparison tests.

**Figure 4 ijms-24-05884-f004:**
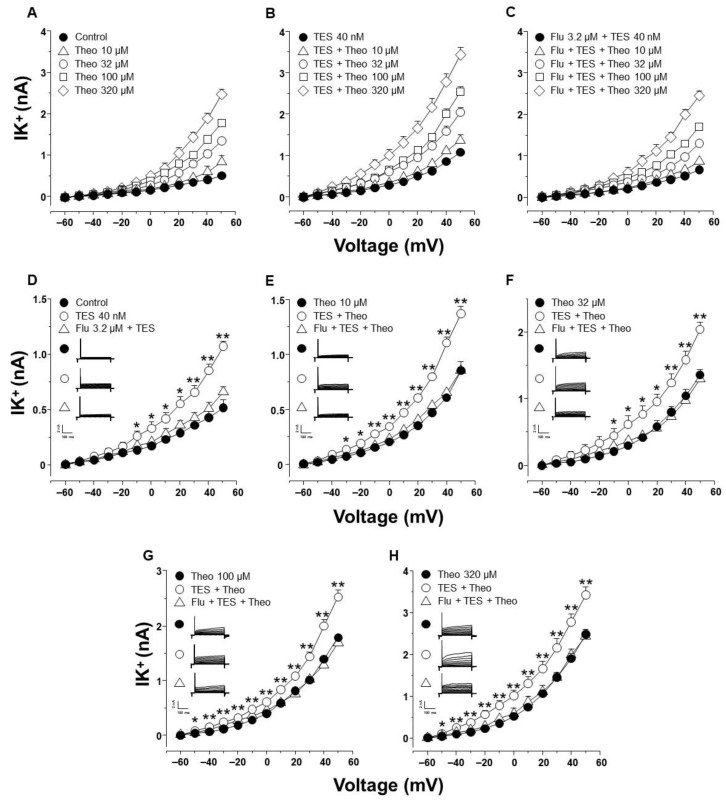
Chronic exposure to testosterone enhances theophylline-induced K^+^ outward currents (IK^+^) via the androgen receptor signaling pathway in cultured tracheal smooth muscle cells. All panels show typical current-voltage curves recorded in single cells. Airway myocytes were subjected to a step depolarization protocol from −60 to +50 mV in 10 mV increments from a holding potential of −60 mV for 500 ms. These stimulations evoked reproducible voltage-dependent K^+^ currents (IK^+^). (**A**) The application of increasing concentrations of theophylline (Theo) to single myocytes elicited a concentration-dependent increase in IK^+^ (*n* = 9). (**B**) Chronic pre-treatment with testosterone (TES, 40 nM for 48 h) enhanced theophylline-induced IK^+^ (*n* = 8). (**C**) The TES-induced increase in IK^+^ was reversed by the androgen receptor antagonist, flutamide (Flu) 3.2 μM (*n* = 7). For illustration, panels (**D**–**H**) summarize the statistical analysis for each concentration of theophylline tested, including experiments with TES alone and flutamide. Note that Flu in these panels abolishes the effect of TES on the theophylline-elicited increase in IK^+^. Note also in (**D**) that TES alone causes a significant improvement in IK^+^ that is abolished by Flu. The insets depict original recordings. Observe that different *Y*-axis scales are used in some figures to make statistical significance clear. Symbols represent mean ± S.E.M. In panel (**D**), * *p* < 0.05, ** *p* < 0.01 when comparing TES (o) or Flu + TES (□) groups vs. control group (●). In panels (**E**–**H**), * *p* < 0.05, ** *p* < 0.01 when comparing TES + Theo group (o) vs. Theo (●) or Flu + TES + Theo (Δ) groups. A one-way analysis of variance was performed, followed by Dunnett’s tests.

**Figure 5 ijms-24-05884-f005:**
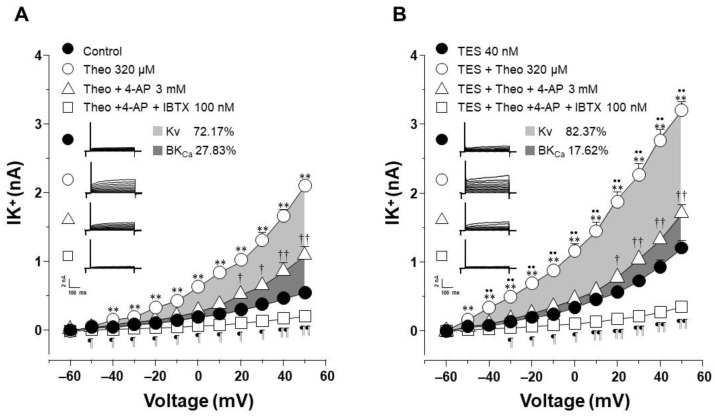
Voltage-dependent delayed rectifier K^+^ (K_V_) channels are involved in the theophylline-induced IK^+^ enhancement after testosterone (TES) chronic incubation in guinea pig tracheal smooth muscle cells. (**A**) Perfusion of theophylline (Theo) 320 μM to single myocytes resulted in a significant increase in IK^+^ from −40 mV ahead, which was partially reduced by 4-aminopyridine (4-AP, 3 mM), a blocker of K_V_ channels. Subsequent administration of iberiotoxin (IBTX,100 nM), a specific blocker of BK_Ca_ channels to the same cells, abolished the remaining theophylline-induced increase in IK^+^ from 20 mV ahead (*n* = 6). Calculation of the area under the curve (AUC) shows that K_V_ channels are responsible for 72.17% of the theophylline response (light gray area), while BK_Ca_ channels account for 27.83% (dark gray area). (**B**) The enhancement of theophylline-evoked increase of IK^+^ by TES from −50 mV ahead, was almost blocked by 4-AP, suggesting a major role of K_V_ channels in the androgen’s action, and the addition of IBTX nearly abolished the IK^+^ (*n* = 6). AUC analysis revealed that the contribution of K_V_ and BK_Ca_ channels to the TES-induced improved theophylline response was 82.37% (light gray area) and 17.62%, (dark gray area), respectively. These results reveal that the main K^+^ channels involved in increasing the IK^+^ triggered by theophylline are the K_V_ channels and suggest that TES upregulates these proteins. Symbols depict the mean ± S.E.M. In panel (**A**), ** *p* < 0.01 when comparing Theo (o) vs. Control (●). † *p* < 0.05, †† *p* < 0.01, comparing Theo + 4-AP group (Δ) vs. Control group (●). ¶ *p* < 0.05, ¶¶ *p* < 0.01, comparing Theo + 4-AP + IBTX (□) vs. Control (●). In panel (B), ** *p* < 0.01 when comparing TES + Theo (o) vs. TES (●). † *p* < 0.05, †† *p* < 0.01, comparing TES + Theo + 4-AP (Δ) vs. TES (●). •• *p* < 0.01, comparing TES + Theo (o) vs. TES + Theo + 4-AP (Δ). ¶ *p* < 0.05, ¶¶ *p* < 0.01, comparing TES + Theo + 4-AP + IBTX (□) vs. TES (●). Repeated measure analyses of variance were performed, followed by Student–Newman–Keuls’ multiple comparison tests.

**Figure 6 ijms-24-05884-f006:**
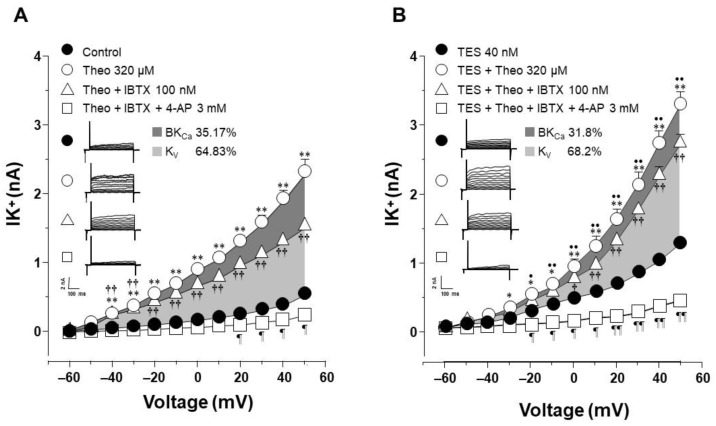
Administration of iberiotoxin followed by 4-aminopyridine shows the same pattern that voltage-gated delayed K^+^ channels (KV) are the major contributors to the enhanced theophylline-induced K^+^ currents after testosterone (TES) treatment. (**A**) Application of theophylline (Theo) 320 μM to single myocytes under control conditions (not incubated with TES) markedly increased the IK^+^ from −50 mV ahead. This increase was slightly reduced by iberiotoxin (IBTX, 100 nM), a specific blocker of BK_Ca_ channels. Successive administration of 4-aminopirydine (4-AP, 3 mM), a blocker of K_V_ channels, to the same cells abolished the remaining Theo-induced increase of IK^+^ from −40 mV ahead (*n* = 6). BK_Ca_ channels account for 35.17% of the theophylline response, according to an estimation of the area under the curve (AUC) (light gray region), while K_V_ channels account for 64.83%. (dark gray area). (**B**) When myocytes were incubated with TES 40 nM, the addition of IBTX (100 nM) reduced the Theo-induced K^+^ current by 31.8% (light gray area), whereas the addition of 4-AP (3 mM) eliminated the theophylline-triggered rise in IK^+^ enhanced by TES, indicating a major role of K_V_ channels (68.2%, dark gray area) in the androgen action (*n* = 6). These results also suggest that TES upregulates these proteins. Symbols depict the mean ± S.E.M. In panel (**A**), ** *p* < 0.01 when comparing Control (●) vs. Theo (o) groups. †† *p* < 0.01, comparing Control (●) vs. Theo + IBTX (Δ). ¶ *p* < 0.05 comparing Control (●) vs. Theo + IBTX + 4-AP (□). In panel (**B**), ** *p* < 0.01 and * *p* < 0.05 when comparing TES (●) vs. TES + Theo (o) groups. † *p* < 0.05, †† *p* < 0.01, comparing TES (●) vs. TES + Theo + IBTX (Δ). •• *p* < 0.01, • *p* < 0.05, comparing TES + Theo (o) vs. TES + Theo + IBTX (Δ) groups. ¶ *p* < 0.05, ¶¶ *p* < 0.01, comparing TES (●) vs. TES + Theo + IBTX + 4-AP (□). Repeated measure analyses of variance were performed, followed by Student–Newman–Keuls’ multiple comparison tests.

**Figure 7 ijms-24-05884-f007:**
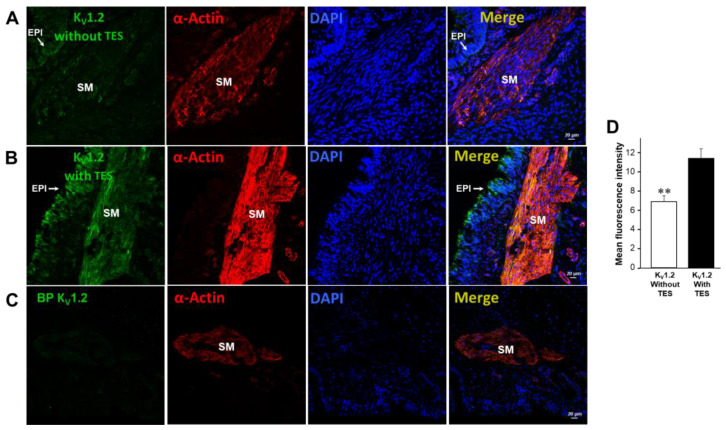
Upregulation of K_V_1.2 in guinea pig airway smooth muscles induced by testosterone (TES) genomic effects is involved in theophylline-induced relaxation enhancement. In the first column, K_V_1.2 immunofluorescence in guinea pig tracheal preparations without (**A**) and with (**B**) 48 h incubation with 40 nM TES is displayed in green on the airway smooth muscle. Note how the fluorescence increased as tissues were exposed to TES. Meanwhile, no fluorescence was observed when the blocking peptide (BP) was incubated together with anti-K_V_1.2 (**C**). The second column shows α-smooth muscle actin (red), and the third column shows nuclei stained with DAPI (blue). The fourth column shows merged images of the first three panels. (**D**) Quantification of K_V_1.2 levels expressed as mean fluorescence, *n* = 5. Error bars represent S.E.M. Significance was determined with an unpaired two-tailed *t*-test. ** *p* < 0.01. EPI = epithelium, SM = smooth muscle.

**Figure 8 ijms-24-05884-f008:**
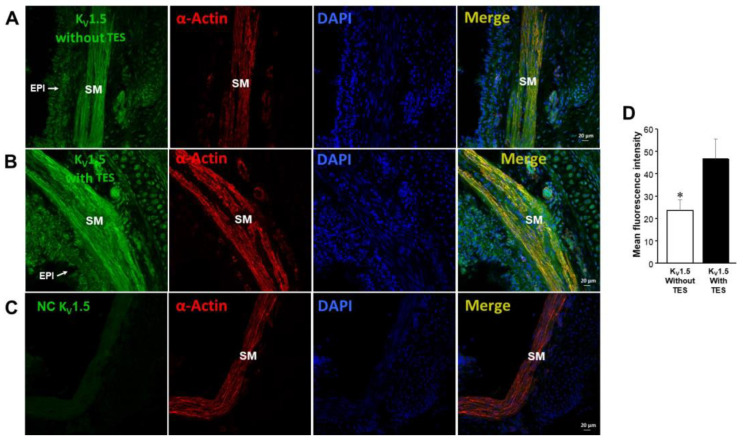
The testosterone (TES)-elicited genomic effect on the enhanced theophylline response is also related to the upregulation of K_V_1.5 in guinea pig airway smooth muscle. The first column shows K_V_1.5 immunofluorescence (green color) in guinea pig tracheal tissues without (**A**) and with (**B**) 40 nM TES exposure for 48 h. Note the increase in fluorescence when tissues were exposed to TES. (**C**) The absence of a primary antibody for K_V_1.5 showed no fluorescence (NC = negative control). The second column shows smooth muscle α-actin (red), and the third column illustrates nuclei (blue). The merged images of the first three panels are shown in the fourth column. (**D**) The bar graph summarizes the quantification of K_V_1.5 levels expressed as mean fluorescence, *n* = 5. S.E.M. is indicated by error bars. Significance was determined using an unpaired two-tailed *t*-test. * *p* < 0.05. SM = muscle, and EPI = epithelium.

**Figure 9 ijms-24-05884-f009:**
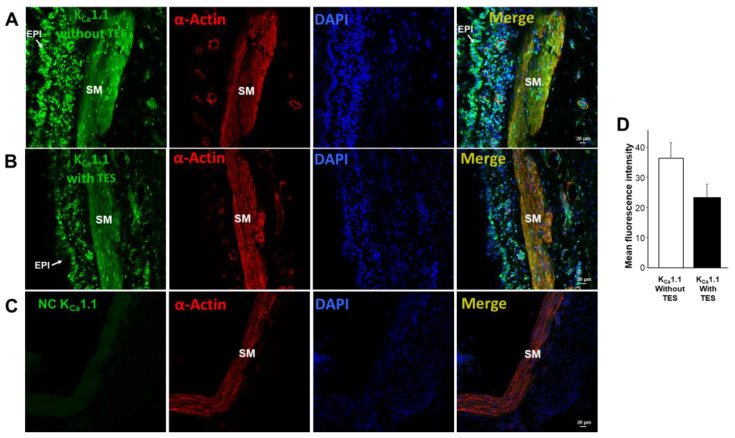
Testosterone fails to increase K_Ca_1.1 levels in guinea pig airway smooth muscle. In the first column, K_Ca_1.1 immunofluorescence (green color) is shown in guinea pig tracheal tissues without (**A**) and with (**B**) 40 nM TES incubation for 48 h. Note that in contrast to K_V_1.2 and K_V_1.5, no increase in fluorescence was observed when tissues were exposed to TES. (**C**) The absence of the primary antibody for K_Ca_1.1 (BK_Ca_ channel) showed no fluorescence (NC = negative control). The second column illustrates smooth muscle α-actin (red), and the third column depicts nuclei stained with DAPI (blue). Merged images of the first three panels are shown in the fourth column. The yellow color indicates the colocalization of K_Ca_1.1 and α-actin. (**D**) Bar graph summarizes the quantification of K_Ca_1.1 levels expressed as mean fluorescence, *n* = 5. S.E.M. is indicated by error bars. SM = smooth muscle, and EPI = epithelium.

## Data Availability

Data are available upon request.

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
