# Peer review of "Theophylline-Induced Relaxation Is Enhanced after Testosterone Treatment via Increased KV1.2 and KV1.5 Protein Expression in Guinea Pig Tracheal Smooth Muscle"

_ijms, 2023, doi:10.3390/ijms24065884_

Round 1

Reviewer 1 Report

The present manuscript demonstrates that pretreatment with testosterone (Tes) enhances the relaxation induced by caffeine, IBMX and theophylline in the isolated guinea pig tracheal ring preparation and the K+ currents, both at basal level and the level augmented by theophylline in a manner sensitive to androgen receptor antagonist flutamide in cultured tracheal smooth muscle cells. The relative contribution of Kv and BK channels to the K current augmented by theophylline was estimated by using their inhibitors. Finally, immunohistochemistry demonstrates increased expression of Kv1.2 and Kv1.5, but not KCa1.1. As a result, authors conclude that testosterone enhances theophylline-induced tracheal relaxation by upregulating Kv1.2 and Kv1.5. 

The study is well designed, the data are sound, the statistical analysis is appropriate, the conclusion is supported by the data, and the manuscript is professionally written. The findings are expected to contribute to understanding gender difference in responsiveness to theophylline for the treatment of asthmatic patients.

Major points

1. It would be recommended to examine the augmenting effect of at least theophylline in female guineapigs. This would reinforce the proposal for gender difference.

2. It is recommended to evaluate the upregulation of Kv1.2 and Kv1.5 with a western blot analysis. Immunofluorescence staining is less quantitative as it depends on the setting of microscopic observation. 

3. The estimation of the relative contribution of  Kv and BK in the K current augmented by theophylline is not readily acceptable. The degree of inhibitory effect of 4-AP and IBTX depends on the order of administration as it was shown that addition of IBTX to 4-AP caused over-inhibition of IK+ above the control level. Please examine the effects of 4-AP and IBTX by applying in an opposite order before concluding the relative contribution. Depending on the observations of the additional experiment, please make appropriate amendment for the estimation of the relative contribution of Kv and BK and the minimal conclusion that can be drawn.

Minor points

1. It is recommended to rephrase the title. There is some syntax error. "by increasing" could be changed "due to" or "via". Alternatively, the title can be started with "Testosterone".

2. Lines 184-185: The statement is recommended to be rephrases as there is syntax error.

3. Fig. 7C is mislabeled as "NC Kv1.2".

4. L.524: Use of pentobarbital as anesthesia is no longer good practice in animal experiment. It could be used as euthanasia and the purpose of its uses in the present study is for euthanasia. "anesthetized" is recommended to rephrase as "euthanized".

5. Lines 572-573: The statement "TES 40 nM or flutaminde 3.2 uM ...... before the addition of TES 40 nM" should be rephrase as there is some confusion.

6. Lines 574-575: "Myocytes spread in the coverslip were ....... allowed to settle down" is recommended to be rephrase, as it appears to be coverslip, but not myocytes, that was placed at the bottom of the chamber and allowed to settle down.

Reviewer 2 Report

The article is deal with theophylline-induced relaxation  that  enhanced after testosterone treatment by increasing protein expression of KV1.2 and KV1.5 3 in guinea pig tracheal smooth muscle . The topic discussed is important for the evaluating gender differences in the effectiveness of asthma drugs.

I would like to make a few comments:

1.      Figure 1:

The word Caffeine write, please, without brackets.

2.      line 157 and further in the text:

In brackets after the numeric values indicate the value p

3.      line 362:

“Because of the androgen´s exposure duration, it seems likely that a genomic mechanism is involved in this improvement.”

The phrase should be changed. One cannot make assumptions about genomic pathways of activation only on the basis of “androgen´s exposure duration”.

4.      Indicate, please, in Materials and methods, what was a gender of guinea pigs.

5.      Did you study the effects on female and male animals?

6.      Discuss the article:

Nowrin U. Chowdhury, Vamsi P. Guntur, Dawn C. Newcomb, Michael E. Wechsler.  Sex and gender in asthma. European Respiratory Review Dec 2021, 30 (162) 210067; DOI: 10.1183/16000617.0067-2021

Author Response

Comments and Suggestions for Authors

The article is deal with theophylline-induced relaxation that enhanced after testosterone treatment by increasing protein expression of KV1.2 and KV1.5 3 in guinea pig tracheal smooth muscle . The topic discussed is important for the evaluating gender differences in the effectiveness of asthma drugs.

I would like to make a few comments:

1. Figure 1:

The word Caffeine write, please, without brackets.

In figure 1, we have deleted the brackets.

  1. line 157 and further in the text:

In brackets after the numeric values indicate the value p

We have added the p value.

  1. line 362:

“Because of the androgen´s exposure duration, it seems likely that a genomic mechanism is involved in this improvement.”

The phrase should be changed. One cannot make assumptions about genomic pathways of activation only on the basis of “androgen´s exposure duration”.

Thank you for your suggestion. These two lines were deleted to avoid misunderstandings.

  1. Indicate, please, in Materials and methods, what was a gender of guinea pigs.

In the original manuscript, line 510, we indicated the gender of the guinea pigs used.

  1. Did you study the effects on female and male animals?

Please refer to the reply to question 1 of Reviewer 1.

  1. Discuss the article:

Nowrin U. Chowdhury, Vamsi P. Guntur, Dawn C. Newcomb, Michael E. Wechsler.  Sex and gender in asthma. European Respiratory Review Dec 2021, 30 (162) 210067; DOI: 10.1183/16000617.0067-2021

We have discussed the manuscript suggested and added some lines in the discussion section (lines 517-518 and 540-543). Also. We have added two new references (89 and 98).

Round 2

Reviewer 1 Report

The manuscript has been satisfactorily revised. 

The results obtained with female ASM are kind of expected results, as authors also mentioned that female cells express androgen receptor as well as male sex hormone. This reviewer does not think that including such results ruin the conclusion and generate confusion, because it could be discussed that the level of endogenous testosterone is a limiting factor for that the observed events take place in female. Inclusion would rather be expected to end up with more accurate, unbiased scientific report. Authors may be welcomed to take a second consideration regarding inclusion of the female data; however, this comment is entirely discretionary.